# Robust extended states in Anderson model on partially disordered random regular graphs

Daniil Kochergin[1,2], Ivan M. Khaymovich[3,4⋆], Olga Valba[2,5] and Alexander Gorsky[2,6]

**1** Moscow Institute of Physics and Technology, Dolgoprudny 141700, Russia
**2** Laboratory of Complex Networks, Center for Neurophysics and Neuromorphic Technologies, Moscow, Russia
**3** Nordita, Stockholm University and KTH Royal Institute of Technology, Hannes Alfvéns väg 12, SE-106 91 Stockholm, Sweden
**4** Institute for Physics of Microstructures, Russian Academy of Sciences, 603950 Nizhny Novgorod, GSP-105, Russia
**5** Higher School of Economics, Moscow, Russia
**6** Institute for Information Transmission Problems, Moscow 127994, Russia

⋆ ivan.khaymovich@gmail.com

## Abstract

In this work we analytically explain the origin of the mobility edge in the partially disordered random regular graphs of degree $d$, i.e., with a fraction $\beta$ of the sites being disordered, while the rest remain clean. It is shown that the mobility edge in the spectrum survives in a certain range of parameters $(d, \beta)$ at infinitely large uniformly distributed disorder. The critical curve separating extended and localized states is derived analytically and confirmed numerically. The duality in the localization properties between the sparse and extremely dense RRG has been found and understood.

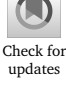

# 1  Introduction

The Anderson model on the Cayley tree allows the analytic derivation of the critical disorder for the localization-delocalization phase transition [1]. More recently, the phase transition in the Anderson model with the diagonal disorder on the hierarchical graphs has found its reincarnation as a toy model for the transition to the many-body localized (MBL) phase in some interacting many-body systems [2]. The simplest ensemble which can be considered as the zeroth approximation to the Hilbert space of the many-body system is the random regular graph (RRG) ensemble [3–32] (see [33] for review).

It was found in [34] that the phase diagram of the Anderson model on RRG, with a finite fraction $\beta < 1$ of disordered nodes, is different from the standard case of $\beta = 1$. In some regions of the $(d, \beta)$-parameter plane, there are delocalized states in the central part of the spectrum that are separated from the localized states by mobility edges at arbitrarily large disorder with the box distribution of the $\beta$ fraction of nodes. This phenomenon takes place if we have some fraction of the clean nodes. Effectively, from the Hilbert-space perspective, there are interacting clean and dirty subsystems in the model.

The physical motivation behind this model is given by the attempt to take into account the topologically protected zero modes in the spectrum of an interacting many-body system [35–38] in the Hilbert-space-graph framework. There are overlaps between these modes and the unprotected modes. Hence, there are links between the clean and dirty nodes in the partially disordered RRG, but this overlap does not destroy their topological nature, hence the corresponding nodes in the RRG are clean. On the other hand, even the coexistence of strongly disordered (MBL) and clean (thermalized) sites in a many-body setting has attracted quite a bit of attention in the literature. [39–43].

In this study, we extend the analysis of [34] and investigate the phase structure of the partially disordered RRG in $(d, \beta)$-parameter space. The region in the $(d, \beta)$ plane where the mobility edge survives at arbitrarily large disorder amplitude will be identified numerically and derived analytically for sparse and extremely dense regimes. The graph-size $N$-dependence of the fractal dimensions $D_q$ and the singular spectrum $f(\alpha)$ for eigenfunctions in the delocalized part of the spectrum is analyzed numerically. We shall explain the microscopic origin of the delocalized eigenstates and identify which aspects of the partially disordered ("two-color") graph architecture, involving the clean and dirty nodes, are crucial for the delocalization.

We shall show that the delocalized states survive when the graph, composed of only clean nodes, has a giant connected component. We also generalize the above approach from the sparse $d \ll N$ to the extremely dense case of the degree $d_c = N - d - 1$. For this, we exploit the duality property between the mobility edges for the partially disordered RRG with the degree $d$ and its complementary counterpart with the degree $(N - d - 1)$. In addition, in the Appendix, we shall investigate the robustness of the above predictions with respect to various perturbations of the RRG. First, we consider the effect of the enhanced number of short cycles, usually almost absent on RRG, on the localization pattern suggested in [22, 44], and second, we investigate the non-Hermitian perturbation of RRG by adding the non-reciprocal directed hopping to the partially disordered RRG as suggested in [45].

Unlike several recent works [46–50], here for the emergence of the mobility edge, robust at the large potential, we need neither the special flat-band structure of the disorder-free model [46–48, 51] nor correlated disorder [49, 50, 52–56]. Our model is based on the i.i.d. disorder potential on the RRG.

The rest of the article is organized as follows. In Section 2 we define the model and present the numerical evidence for the mobility edge at arbitrarily large disorder. In Section 3 we analytically derive the critical curve in $(d, \beta)$-parameter space for the mobility edge. In Section 4 we generalize our analytical consideration to the extremely dense graphs by analytically utilizing the duality between the localization patterns for node degrees $d$ and $(N - d - 1)$ and confirming the results numerically. Section 5 concludes the results. In Appendices we consider the multifractal spectrum and prove the robustness of the phenomena observed with respect to the small perturbations of RRG by non-Hermitian deformation and the enhanced number of the short cycles, usually almost absent on RRG.

## 2 Robustness of delocalization in partially disordered RRG

In this Section we consider the numerical simulation of the RRG, with the fraction $\beta$ of sites subject to the disorder of i.i.d. random variables $\epsilon_i$ of the amplitude $W/2$ taken from the uniform distribution, $|\epsilon_i| < W/2$. First, in Sec. 2.1 we introduce the model, and second, in Sec. 2.2 we present the numerical simulations for the spectral and localization properties of the model across the spectrum.

### 2.1 The model

In the conventional framework, one studies the Anderson transition for non-interacting spinless fermions hopping over RRG with the connectivity $d = 3$ in a diagonal disorder described by Hamiltonian

$$H = \sum_{i,j} A_{ij} \left( c_i^+ c_j + c_i c_j^+ \right) + \sum_{i=1}^{\beta N} \epsilon_i c_i^+ c_i. \tag{1}$$

The first sum, representing the hopping between nearest-neighbor RRG nodes $i$ and $j$, is written in terms of the adjacency matrix ($A_{ij} = 1$ for nearest neighbors and $A_{ij} = 0$ otherwise) for the regular graph, $\sum_i A_{ij} = \sum_j A_{ij} = d$. The second sum, running over all $N$ nodes, represents the potential disorder. The standard fully disordered RRG ensemble, corresponding to $\beta = 1$, undergoes the Anderson localization transition at $W_c = 18.16$ for $d = 3$ [8, 11, 15, 33]. For larger $d$ the critical disorder is usually estimated as

$$W_c(d) \simeq d \ln d. \tag{2}$$

### 2.2 Robustness of delocalization and fractal dimension $D_2$

Let us investigate numerically the properties of the states in the delocalized spectral part, found in [34] in the large $W$ limit. As the probes, we choose the density of delocalized states, $\rho(E) = \left\langle \sum_{n \in \text{delocalized}} \delta(E - E_n) \right\rangle$, the spectral level-spacing statistics, $P(s)$, with $s_n = E_{n+1} - E_n$, and the dependence of the fractal dimension $D_2(N) \equiv -\ln \left( \sum_i |\psi_E(i)|^4 \right) / \ln N$ of an eigenstate $\psi_E(i)$ on the point in the $(d, \beta)$ parameter plane.

First, let us demonstrate that the delocalized states survive at a very large disorder and are clearly seen numerically. Figure 1 clearly demonstrates that at large $W$, the width of the delocalized energy range is $W$-independent. An additional level, started at $E = d$ at $W = 0$, corresponds to the standard eigenstate of the adjacency matrix, which is homogeneous over the entire graph. Its separation from the bulk bandwidth at $W \lesssim 2d$ protects it from most of

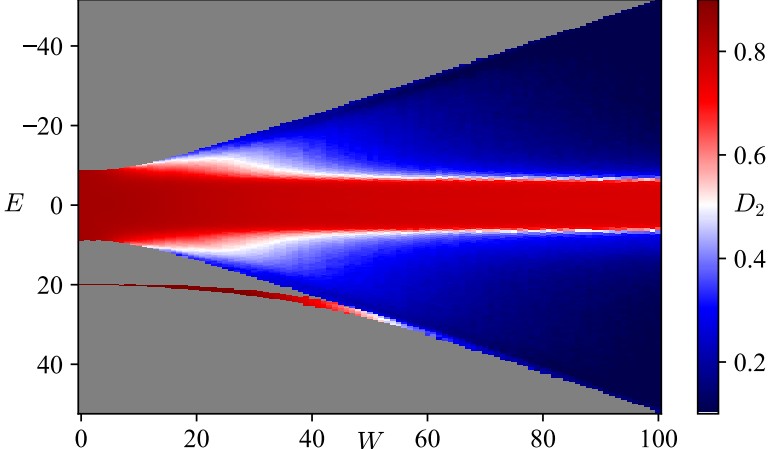

Figure 1: **Mobility edge structure versus disorder $W$**. The color plot shows the fractal dimension $D_2$ versus the disorder $W$ and eigenvalues $E$ in the partially disordered RRG of size $N = 1024$, with connectivity $d = 20$, and the fraction of disordered nodes $\beta = 0.5$. The data is averaged over 100 realizations. Figure shows that delocalized states survive at any achievable disorder amplitude $W$.

the localization mechanisms. At larger disorder amplitudes, it merges with the bulk spectrum and then localizes.

Note that here and further, we focus mostly on the localization, $D_2 = 0$, versus the delocalization, $D_2 > 0$, but not on the ergodicity, $D_2 = 1$, versus the non-ergodicity, $0 < D_2 < 1$. Already in a fully disordered RRG at $\beta = 1$, the question of the existence of a non-ergodic phase in RRG has been a discussion point for years [4–20, 27–32], and even now the maximal system sizes of a few millions, $N \sim 10^6$, do not resolve this issue [14, 31, 32]. Therefore, in this work we calculate the fractal dimensions $D_2$ (and their generalization $D_q$ together with the singularity spectrum $f(\alpha)$ with the definitions given below) in the Appendix A only of finite sizes up to $N \sim 30000$ and do not claim any ergodicity or non-ergodicity. In addition, we have checked that the above picture of the mobility edge, see Fig. 1, converges with the system size much below the maximal considered size of $N \sim 30000$.

Figure 2 demonstrates finite-size data, $D_2(N)$, up to $N \sim 30000$ and its infinite-size extrapolation, $D_2(N) = D_2 + c_2/\ln N$, with a certain fitting parameter $c_2$ (see [8, 57, 58] for more details on extrapolation), for partially disordered RRG with connectivity $d = 3$ and the fraction of disordered nodes $\beta = 0.5$ at intermediate disorder amplitudes $W = 30$. The mobility edges calculated from (18) stay at the same energy regardless of the system size, while the fractal dimension flows upwards between the mobility edges and goes to zero beyond it.

The delocalization can also be checked via the level spacing distribution $P(s)$, see Fig. 3. Level spacing determines the statistics of spacing between two adjacent energy levels $s_i = E_{i+1}^u - E_i^u$, where $E_i^u$ are energy levels after the unfolding procedure (see, e.g., [26] for details). There, the eigenenergy statistics show the standard repulsion inside the delocalized region and the Poisson statistics beyond the mobility edge [59]. Some deviations from the Poisson statistics for the localized nodes, $|E| > E_{ME}$, should be related to the small DOS for these states and its fluctuations for large $W = 1000$, see the further discussion of Fig. 4 below.

The density of states, $\rho(E)$, see Figs. 2(b) and 4, shows a clear separation into two parts: the states, localized at disordered nodes, form a flat box-like distribution of the width $W \gg d$ (barely seen in Fig. 4), while the extended ones are confined at small energies, $|E| \lesssim 2\sqrt{(1-\beta)(d-1)}$. At small $\beta$, the density of delocalized states, $\rho(E)$, is close to the

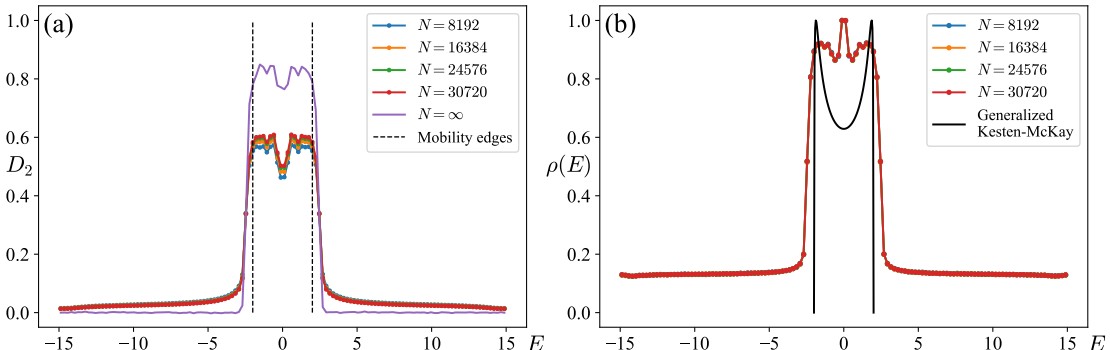

Figure 2: **(a) Fractal dimension $D_2$ and (b) density of states across the spectrum on partially disordered RRG** at several sizes $N$, with connectivity $d = 3$, disorder amplitude $W = 30$ and fraction of disordered nodes $\beta = 0.5$. Colored symbols show finite-size data, while the solid purple line shows an extrapolated curve [8, 57, 58]. Black dashed lines in (a) show the mobility edges, calculated from (18). The black solid line in (b) shows a generalized Kesten-McKay distribution of delocalized states, calculated from (17). The flat contribution of the localized states is clearly seen in (b).

Kesten-McKay distribution [60, 61]

$$\rho(E) = \rho_{KM}(E) = \frac{d\sqrt{[4(d-1) - E^2]}}{2\pi(d^2 - E^2)}, \tag{3}$$

while at large $\beta$ it becomes close to the Wigner-Dyson distribution (Fig. 4). We confirm this behavior later in Eq. (17) by the analytical consideration. It is expected since at a small $\beta$, the clean nodes form almost RRG, while at larger $\beta$, the clean-node graph gets randomized by the dirty nodes.

As the localized states live mostly on the dirty nodes, they are subject to the box-distributed disorder of the amplitude $W = 1000$. The number of such localized states is $\beta \cdot N \simeq 300 - 500$ in Fig. 4. As a result, in normalized DOS, $\int \rho(E)\, dE = 1$, shown in Fig. 4, the contribution of such localized states is rather small, $\rho(|E| < W/2) \simeq \beta/W \simeq 0.0001 - 0.0007$ on average in panels (a)-(d). At small $\beta$, panel (a), in one realization of the graph only a few localized states, $\sim \beta \Delta E N/W \simeq 0.6$, appear in the shown interval $|E| < \Delta E = 6$ and this gives barely seen fluctuations (do not associate a peak at $E \simeq d$ with them). With increasing $\beta$ from panel (a) to (d) the number of localized states grows, and so does the background, becoming more and more homogeneous and close to the box distribution of the diagonal disorder. At intermediate $W \simeq 30$, shown in Fig. 2(b), this box contribution to the DOS overcomes the threshold of the noise.

In the central part of the spectrum, the deviations from the predicted behavior, Eq. (17), are expected in Fig. 4 as the parameter $(1-\beta)d$ goes down, making our cavity-method approximation Eqs. (11) and (12) less and less accurate. Other deviations, see, e.g., Fig. 2(b) come from the corrections in the small parameter $d/W$, neglected in the analytical consideration for simplicity.

In Fig. 2(b), the black line shows only the partial DOS of the delocalized states at $W \to \infty$. Therefore, even without finite-$W$ corrections, one expects to see the overall shift of the DOS due to the localized state contribution (which is of the order of 0.15 in that figure). This explains the overall upward shift of the data with respect to the analytical black curve. The peaks around $|E| \lesssim 2\sqrt{(d-1)(1-\beta)}$ in the analytical formula (17) are related to the smallness of $d = 3$ and therefore to the closeness in Eq. (17) of the spectral edges $\pm 2\sqrt{(d-1)(1-\beta)}$ of

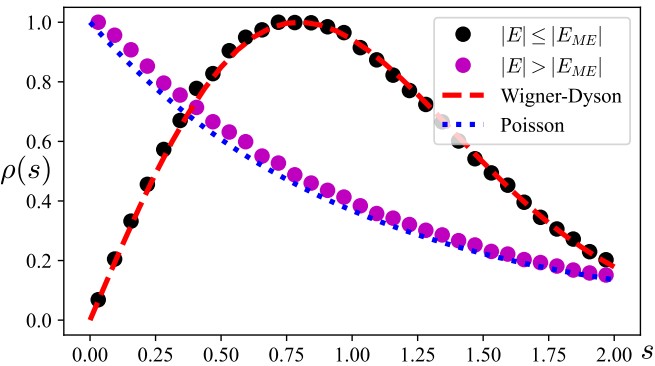

Figure 3: **Level spacing distribution between (black dots) and beyond (purple dots) the mobility edges in the partially disordered RRG** of the size $N = 8192$, with the connectivity $d = 10$ and the fraction of disordered nodes $\beta = 0.5$ at the disorder $W = 1000$. The data is averaged over 1024 realizations. The data within (beyond) the mobility edge is well described by red dashed (blue dotted) line, corresponding to the Wigner-Dyson (Poisson) distribution.

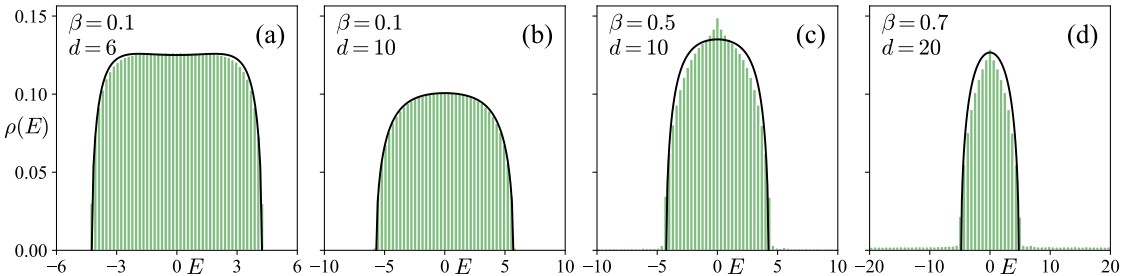

Figure 4: **Density of states of partially disordered RRG for different node degrees $d$ and fractions of disordered nodes $\beta$.** Green-colored histograms show numerically calculated spectral densities for the size $N = 8192$ and disorder amplitude $W = 1000$. The data is averaged over 100 realizations. Black lines show generalized Kesten-McKay distributions of delocalized states calculated from (17) for each panel. (a) $\beta = 0.1$, $d = 6$, (b) $\beta = 0.1$, $d = 10$, (c) $\beta = 0.5$, $d = 10$, (d) $\beta = 0.7$, $d = 20$. The contribution of the localized states is barely seen at such large disorder.

the delocalized states and the divergences in DOS $\pm d\sqrt{(1-\beta)}$ to each other. The rounding of these peaks in the data might be finite-size-$N$ and finite-disorder-$W$ effects.

The width of the delocalized energy range has nontrivial $(d, \beta)$ dependence, see Fig. 5(a). There, the above-mentioned fluctuations in DOS from the localized nodes, box-distributed with the width $W = 1000$, have been eliminated by putting a threshold to the DOS data, see the caption of Fig. 5.

There is the critical curve $\beta_c(d)$ in the parameter space that separates the regime with and without the mobility edge, see Fig. 5(c). This is related to the percolation via the clean nodes on the partially disordered RRG, see the analytical consideration in the next section.

We have also investigated the $N$-dependence of the fractal dimension

$$D_q \equiv \frac{\ln\left(\sum_i |\psi_E(i)|^{2q}\right)}{(1-q)\ln N}, \tag{4}$$

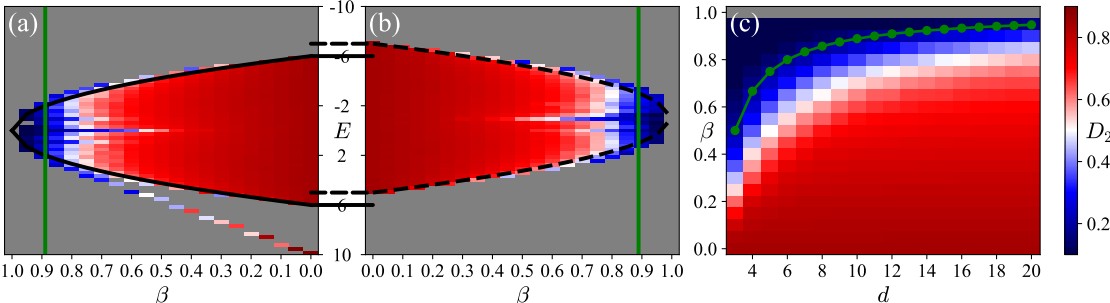

Figure 5: **Mobility edge structure versus the fraction of disordered nodes $\beta$ and the connectivity $d$.** Color plots in panels (a) and (b) show the fractal dimension $D_2$ versus $\beta$ and eigenvalues $E$ in the partially disordered RRG of size $N = 1024$ at the disorder amplitude $W = 1000$ for (a) the dilute graph with the connectivity $d = 10$ and (b) the extremely dense graph with the connectivity $d_c = N - d - 1 \simeq N$, which is complement to the one in (a). Black solid (dashed) line in (a) [(b)] denotes the mobility edge, $|E_{ME}|^2 = 4(1 - \beta)(d - 1)$, Eq. (18), ($|E_{ME} + 1|^2 = 4(1 - \beta)(d - 1)$) for the sparse (dense) graph. The localized states are spread over a huge energy interval $|E| < W/2 = 500$ and thus, give only a noise-like contribution to DOS. In order to make the data smooth and accessible, we have put a threshold on the DOS and got rid of this noisy part. The grey color in panels (a) and (b) corresponds to the DOS below the threshold. (c) Color plot of the average fractal dimensions $D_2$ over the states in the central band $|E| < E_{ME}$ in the partially disordered RRG versus $\beta$ and $d$ at $W = 1000$. Green solid lines in all panels show the critical curve $\beta_c$, given by (19). All the data is averaged over 25 realizations.

and spectrum of fractal dimensions

$$f(\alpha) \equiv 1 + \frac{\ln P\left(\alpha = -\frac{\ln|\psi_E(i)|^2}{\ln N}\right)}{\ln N} . \tag{5}$$

The corresponding plots for $\beta = 0.5$ and $\beta = 0.75$ are presented in Fig. 7 − 10 in Appendix A. The peak in the central part of the DOS and the corresponding deviations in the fractal dimensions in Fig. 2(a) and Fig. 4(c, d) (it is not always a dip in $D_q$) is considered in detail numerically in the multifractal analysis in Appendix A, where this DOS peak is called as the central part of the spectrum. From the previous literature (see, e.g., [44]), we know that the dips in $D_q$ at certain energies can be related to the isolated small clusters of clean nodes or to so-called topologically equivalent nodes.

## 3 Derivation of critical curve at $(d, \beta)$ plane

In this section, we explain why the density of the delocalized states at not very large $\beta$ is well-approximated by the Kesten-McKay distribution with the rescaled RRG $d^*$ and tree $d_t^*$ degrees. This rescaling reproduces correctly the numerical result for the spectral width of the delocalized range and the critical curve at the $(d, \beta)$ plane at relatively small $d \ll N/2$. Note that the one-loop correction for the Kesten-McKay law has been found in [62] and a more general cavity analytic approach for dense graphs has been developed in [63].

This result can be straightforwardly understood as follows. For large enough disorder $W \gg 1, d$, all the dirty nodes of the RRG become localized, and the only possibility for extended states to survive is to live exclusively on the clean nodes. This reduces the problem to the RRG

with the $\beta$ fraction of edges being removed from that. This graph should be equivalent to the Erdös-Rényi one or other hierarchical graphs with fluctuating connectivity [64] with a certain distribution of the number of edges.

In order to make the above argument clear, as on the usual RRG, let us consider cavity equations for the single-site Green's functions on clean $G_n^*$ and dirty $G_n'$ nodes and their tree counterparts $G_{n\to a}^*$ and $G_{n\to a}'$ with the removed link from $n$ to its ancestor $a$.

$$
\begin{cases}
\frac{1}{G_n^*} = E + i\eta - \sum_{m=1}^{k_n} G_{i_m^*\to n}^* - \sum_{m=1}^{d-k_n} G_{i_m'\to n}', \\[2mm]
\frac{1}{G_n'} = E + i\eta - \varepsilon_i - \sum_{m=1}^{k_n} G_{i_m^*\to n}^* - \sum_{m=1}^{d-k_n} G_{i_m'\to n}', \\[2mm]
\frac{1}{G_{n\to a}^*} = E + i\eta - \sum_{m=1}^{l_n} G_{i_m^*\to n}^* - \sum_{m=1}^{d-1-l_n} G_{i_m'\to n}', \\[2mm]
\frac{1}{G_{n\to a}'} = E + i\eta - \varepsilon_i - \sum_{m=1}^{l_n} G_{i_m^*\to n}^* - \sum_{m=1}^{d-1-k_n} G_{i_m'\to n}',
\end{cases}
\tag{6}
$$

where $i_m^*$ and $i_m'$ are the indices, enumerating the pure and disordered sites on the tree, the ancestor of which is $n$, $k_n$ and $l_n$ are numbers of clean descendants of $n$ on the RRG ($G_n$) and on the tree ($G_{n\to a}$), respectively. It is important to note that the total number of the descendants of $n$ for the tree is given by a branching number $d_t = d - 1$, while for the RRG, where each point is locally a root of the tree, it is given by the vertex degree $d$.

The number of clean nearest descendants of any node $n$ obeys the binomial distribution

$$
p_{\tilde{d}}(k) = \binom{\tilde{d}}{k}(1-\beta)^k \beta^{\tilde{d}-k},
\tag{7}
$$

with $\tilde{d} = d_t$, $k = l_n$ for the tree ($p_{d_t}(l_n)$) and $\tilde{d} = d$, $k = k_n$ for the RRG ($p_d(k_n)$). In both cases, for large enough $(1-\beta)\tilde{d} \gg 1$ this distribution is well approximated by the normal distribution with the mean and the variance given by

$$
\langle k \rangle_{\tilde{d}} = \sum_k p_{\tilde{d}}(k)k = (1-\beta)\tilde{d},
\tag{8}
$$

$$
\sigma_{\tilde{d}}^2 = \sum_k p_{\tilde{d}}(k)(k - \langle k \rangle_{\tilde{d}})^2 = \beta(1-\beta)\tilde{d}.
\tag{9}
$$

Let us consider the simplest approximation at large $W$ by keeping in the equation for the dirty nodes only the disorder term which yields

$$
G_n' \propto W^{-1},
\tag{10}
$$

and substitute this solution into the equation for the clean nodes. In the limit $W \to \infty$, the effects of dirty nodes are subleading, and the problem reduces to the one on the disorder-free nodes on a graph with node degree distribution (7). Hence we get the equation for clean nodes

$$
\frac{1}{G_n^*} = E + i\eta - \sum_{m=1}^{k_n} G_{i_m^*\to n}^*,
\tag{11}
$$

$$
\frac{1}{G_{n\to a}^*} = E + i\eta - \sum_{m=1}^{l_n} G_{i_m^*\to n}^*.
\tag{12}
$$

These equations evidently yield the RRG KM spectral density, but now both with fluctuating and rescaled $d^* = k_n$ and $d_t^* = l_n$. For large enough $\langle k \rangle_d \gg 1$ the corresponding rescaled parameters in the most realizations are given by their mean values

$$d^* = \langle k \rangle_d = (1-\beta)d, \quad d_t^* = \langle l \rangle_{d_t} = (1-\beta)d_t, \tag{13}$$

and their relative fluctuations are small as $\sigma_d/d^* \sim \sqrt{\beta/d^*} \ll \beta^{1/2} \leq 1$ for large $d^*$ in our consideration. In addition to that, following the results of [65–67], one concludes that the effects of fluctuations in graph degree are important for the eigenstate localization properties as soon as the variance (but not the standard deviation) is large compared to the mean value. This is not the case in our limit as well $\sigma_d^2/d^* = \beta \lesssim 1$. The critical value $\beta_c$, when the clean nodes do not form a connected tree-like graph, can be derived from the equation $d_t^*(\beta_c) = 1$.

Note that, unlike the regular case, both rescaled parameters $d^*$ and $d_t^*$ are *not* anymore related to each other via $d^* = d_t^* + 1$ (similarly to [64]).

The generalized KM distribution can be obtained from Eqs. (11) and (12). In the limit $(1-\beta)\tilde{d} \gg 1$, when due to the large effective connectivity of clean nodes, it is natural to assume that $G_{n \to a}^*$ is self-averaging, one can rewrite the latter of two equations as a self-consistent equation on the mean $\langle G_{n \to a}^* \rangle = G_\to^*$ as follows

$$\frac{1}{G_\to^*} = E + i\eta - d_t^* G_\to^*, \tag{14}$$

which immediately gives the solution

$$G_\to^* = \frac{E + i\sqrt{4d_t^* - E^2}}{2d_t^*}, \quad d_t^* = (1-\beta)(d-1). \tag{15}$$

with the semi-circular density of states $\rho_\to = \mathrm{Im}\, G_\to^*/\pi$.

The generalized KM distribution is given by the equation (11) for $\langle G_n^* \rangle = G^*$ with $k_n \simeq d^* = (1-\beta)d$

$$G^* = \frac{1}{E + i\eta - d^* G_\to^*} = \frac{(d-2)E + id\sqrt{4d_t^* - E^2}}{2[d^2(1-\beta) - E^2]}. \tag{16}$$

This gives for the density of states $\rho$

$$\rho(E) = \frac{\mathrm{Im}\, G^*}{\pi} = \frac{d\sqrt{4(d-1)(1-\beta) - E^2}}{2\pi[d^2(1-\beta) - E^2]}, \tag{17}$$

and the corresponding mobility edges $E_{ME}$ at

$$E_{ME} = \pm\sqrt{4(d-1)(1-\beta)}. \tag{18}$$

Critical value $\beta_c$ can be determined by the existence of a giant component in the graph of clean nodes. For arbitrary node degree distribution, the graph almost surely has a giant component if $\langle k^2 \rangle - 2\langle k \rangle > 0$ [68]. For distribution $p_d(k)$ (7) critical value $\beta_c$ is

$$\beta_c(1-\beta_c)d + (1-\beta_c)^2 d^2 - 2(1-\beta_c)d = 0 \;\to\; \beta_c = 1 - \frac{1}{d-1}. \tag{19}$$

Note that one can derive the same result from the standard KM density of states: the critical value $\beta_c$ is defined as the percolation threshold $d_t^* = 1$ on the tree with the branching number $d_t^*$, Eq. (15), as the formation of the giant connected component is described by the equation (14) on the corresponding tree. If $\beta \leq \beta_c$, the graph of clean nodes has a giant connected component, and the wave functions on this component are delocalized. If $\beta > \beta_c$, the graph

of clean nodes separates into disconnected components, the average size $n$ of each of those is small compared to the network size, $n \ll N$. Localized eigenstates in Fig. 5(a) significantly below threshold appear due to the isolated pure nodes at $\lambda = 0$ and connected pairs of pure nodes at $\lambda = \pm 1$. Probably, it is these isolated clean nodes that lead to the deviations of DOS from Eq. (17) in Figs. 2(b) and 4(c), (d).

Note that the above problem might be similar to the one of the Erdös-Rényi graph, where some states can be localized even without disorder due to the fluctuating extensive node degree $d \sim (\ln N)^a$, $1 < a < 2$ [65–67]. Different approximation schemes were proposed for the analytically calculated density of states [69, 70].

The robustness of the delocalized states with respect to various perturbations, suggested in the literature [44, 45], is considered in Appendix B. There we focus on the non-Hermitian versions of RRG with the (partially) directed edges (see Appendix B.1), as well as the presence of the short cycles of length 3, which are usually almost absent in the RRG. The increase of the short-cycle number is achieved by a certain deformation of the distribution over all possible RRG by adding an exponential weight of the number of such cycles [22, 44], see Appendix B.2.

Both generalizations show that small perturbations do not break the presence of the extended states below the mobility edge and confirm the robustness of the above conclusions.

## 4  Duality in localization properties between sparse and dense RRG

The analytical derivations of the density of states for the delocalized states, Eq. (17), and the corresponding mobility-edge location, Eq. (18), should be valid for large enough disorder amplitudes $W \gg 1, d$ and effective degrees of the graph of clean nodes, $d_t^*, d^* \gg 1$, Eq. (13), but for the corresponding values of the total vertex degree $d$ of the graph, not limited from above.

However, the numerical simulations in Fig. 5(b) show that this is not the case for the dense RRG at large $d$, when $|N - d| \ll N$. In this case, the energy interval, where the states are delocalized and the mobility edge curve in the $(d, \beta)$-plane exists, is not determined by the large degree $d$, but instead by the node degree of the complementary graph, $d_c = N - d - 1 \ll N$. Indeed, the comparison of Fig. 5(a) and (b) shows that the width of this interval $\Delta E$ is

$$\Delta E = (1 - \beta) \min[d, (N - d - 1)] - 1, \tag{20}$$

that corresponds to the results of the complementary graph with $d_c = N - d - 1 \ll N$.

For the adjacency matrix, consisting of 0 and 1, and for the symmetric disorder distribution, the above mapping to the complementary graph can be straightforwardly understood via the rank-1 perturbation of the initial problem, see [71].

Indeed, using the eigenvalues $E_n^0$ and eigenvectors $\left| E_n^0 \right\rangle$ of a certain realization of the problem on the standard (complementary) graph with the connectivity $d_c \ll N$ and the diagonal disorder $\varepsilon_i$, well-described by Eqs. (17) and (18), one can straightforwardly write the Hamiltonian of the dense model (with $d = N - d_c \simeq N$) as a complementary graph as follows

$$H = -\sum_n E_n^0 \left| E_n^0 \right\rangle \left\langle E_n^0 \right| + |1\rangle \langle 1| - I. \tag{21}$$

Here $\langle i | 1 \rangle = 1$ for all sites $i$ and $I$ is the identity matrix, as the vector $|1\rangle$ of ones is not normalized. Note that for any disorder realization on the initial (complementary) graph $\epsilon_{i,c}$, the effective disorder realization in the dense one changes its sign $\epsilon_i = -\epsilon_{i,c}$.

The peculiar property of the model with a large connectivity $d \simeq N$ is that the part $|1\rangle \langle 1|$, non-diagonal in the eigenstate basis of the initial sparse problem $\{\left| E_n^0 \right\rangle\}$, is a rank-1 matrix,

and therefore this dense model can be diagonalized using the simplest Bethe ansatz solution of the Richardson's model [72–76].

$$\sum_n \frac{|\langle E_n^0|1\rangle|^2}{E + E_n^0 + 1} = 1 \,, \tag{22}$$

$$|E\rangle = C_E \sum_n \frac{\langle E_n^0|1\rangle}{E + E_n^0 + 1} |E_n^0\rangle \,, \tag{23}$$

$$C_E^{-2} = \sum_n \frac{|\langle E_n^0|1\rangle|^2}{(E + E_n^0 + 1)^2} \,. \tag{24}$$

From the literature [58, 75, 77–79] it is known that, as soon as $\left|\langle E_n^0|1\rangle\right|^2$ is more or less homogeneous versus $n$ and $W \ll N$, all but one new eigenvalue, being the solutions of Eq. (22), $E = E_n$ (shifted by 1 in our case due to the presence of $I$ in the equation) are located in between the old ones $-E_{N-n+1}^0 < E_n + 1 < -E_{N-n}^0$ and the eigenstates are power-law localized in the eigenbasis of the initial problem with the power-law exponent 2, $|\langle E_{N-n}^0|E_m\rangle|^2 \sim 1/|m-n|^2$. This property is related to the fact that for $W \ll d \sim N$ all but one of the eigenstates are nearly orthogonal to $|1\rangle$.

The only high-energy level (not shown in Fig. 5(b)), which is not orthogonal to $|1\rangle$, takes the large energy of the order of $E_N \sim N$. As soon as the diagonal disorder $W \ll N$, this vector is delocalized as $|E_N\rangle \simeq |1\rangle / \sqrt{N}$.

This immediately means that

- The width and the profile of the band are the same in the initial and complementary problems and controlled by the effective node degree $d_{\text{eff}} = \min(d, N - d - 1)$;

- The localization and fractal properties $D_q$ are also the same, at least for $q > 1/2$, where the power-law localization tails are not important;

- The only difference appears at $W \ll N$, when there is the high-energy level at $E_N \simeq N$, while the bulk bandwidth is shifted by $-1$ (due to the same trace of both initial and complementary matrices).

All these properties have been numerically investigated in Fig. 5(b), please compare with the panel (a) to see the shift of energy by 1.

## 5 Conclusion

In this study, we have clarified the mechanism behind the robustness of the delocalized energy range at arbitrarily large disorder, found in [34]. The system involves coupled clean and dirty subsystems, and the delocalized region at the $(d, \beta)$-parameter plane corresponds to an effective problem solely on the clean nodes, with the renormalized RRG and tree degrees, at the large enough disorder. This result has been obtained analytically in the leading approximation in $1/W$ and at large but finite node degree, and confirmed numerically for sparse and extremely dense regimes. In addition, in Appendices, the effects of various perturbations of $\beta$-deformed RRG have been investigated as well.

The pattern of the appearance of the controllable mobility edge we have found provides additional insights for the account of the topologically protected modes of the interacting many-body systems in the Hilbert space framework. In this respect it would be of particular interest to generalize the effect of $\beta$-deformation to the many-body Hilbert space structures, like a hypercube graph in the quantum random energy model [80, 81]. It is also interesting

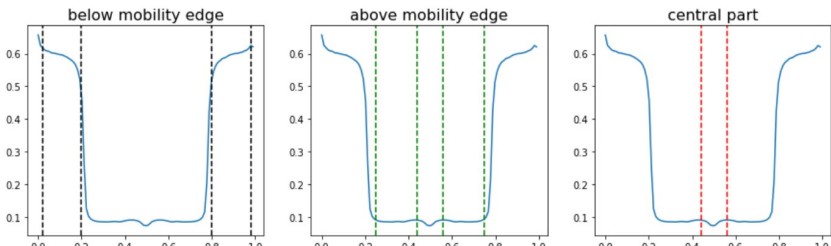

Figure 6: **The finite-size approximation of the fractal dimension,** $D_q(N) = \ln[IPR(N)/IPR(2N)]/\ln[2]$, **versus energy** for $W = 30$, $d = 3$, and $\beta = 0.5$, used to separate the energy windows for the next four figures: (left) below the mobility edge (localized states), $|E| > E_{ME1}$; (middle) above the mobility edge (delocalized states), $E_{ME2} \leq |E| \leq E_{ME1}$; (right) in the central (not fully ergodic) part, $|E| < E_{ME2}$.

to consider a randomly distributed $\beta$ parameter and the effects of the non-Hermitian diagonal disorder, which may lead to the localization enhancement [82–85], unlike the usual non-reciprocity [86]. If the RRG ensemble is considered as the discrete model for the 2d quantum gravity, the Anderson model corresponds to the massive field coupled to the fluctuating geometry. The case of the Anderson model on partially disordered RRG corresponds to the situation when there are zero modes of the field localized at some defects. It would be interesting to develop this framework further.

## Acknowledgments

We thank A. Scardicchio for fruitful discussions. A.G. thanks Nordita and IHES where the parts of this work have been done for the hospitality and support.

**Funding information** I. M. K. acknowledges the support by the Russian Science Foundation, Grant No. 21-12-00409.

## A  Multifractal spectrum $f(\alpha)$ and fractal dimensions $D_q$

In this Appendix, we show the multifractal analysis for the spectrum of fractal dimensions, Figs. 7 and 8, and for the fractal dimensions, Figs. 9 and 10, on the RRG with $d = 3$ for two different values of $\beta$ in the three distinct part of the spectrum, shown in Fig. 6.

For $\beta = 0.5$, smaller than a threshold value, Eq. (19), see Figs. 7, 9, the states in the bulk part of the spectrum, $|E| < E_{ME2}$ are delocalized at any available disorder amplitude ($f(0)$ stays significantly negative and $D_q > 0$). Unlike this, above the threshold value, $\beta = 0.75 > \beta_c$, see Figs. 8, 10, all the states tend to the localization eventually at large enough disorder. This confirms the main claims of the main text.

In addition, one can see some deviations from ergodicity in the delocalized parts (two rightmost rows in Figs. 7, 9), that may though be finite-size effects. Therefore, in the main text we don't claim any fractality or multifractality of these states, focusing on the localization (leftmost rows) versus the delocalization (the rest).



Figure 7: **The spectrum of fractal dimensions $f(\alpha)$ for different disorder amplitudes $W$, $d = 3$, and $\beta = 0.5$ in different parts of spectrum** (see Fig. 6): (left) below the mobility edge (localized states), $|E| > E_{ME1}$; (middle) above the mobility edge (delocalized states), $E_{ME2} \leq |E| \leq E_{ME1}$; (right) in the central (not fully ergodic) part, $|E| < E_{ME2}$. Colored symbols show finite-size data, while the solid purple line shows an extrapolated curve [8,57,58]. Panels show gradual localization of the states below the mobility edge (left) with increasing disorder $W$ ($f(0)$ goes to 0), while both states above it and at the central part stay delocalized ($f(0)$ stays significantly negative).



Figure 8: **The spectrum of fractal dimensions $f(\alpha)$ for different disorder amplitudes $W$, $d = 3$, and $\beta = 0.75$ in different parts of spectrum** (see Fig. 6): (left) below the mobility edge (localized states), $|E| > E_{ME1}$; (middle) above the mobility edge (delocalized states), $E_{ME2} \leq |E| \leq E_{ME1}$; (right) in the central (not fully ergodic) part, $|E| < E_{ME2}$. Colored symbols show finite-size data, while the solid purple line shows an extrapolated curve [8,57,58]. Panels show gradual localization of all the states with increasing disorder $W$, both below the mobility edge, above it, and at the central part as $\beta > \beta_c = 1 - (d-1)^{-1}$, Eq. (19).

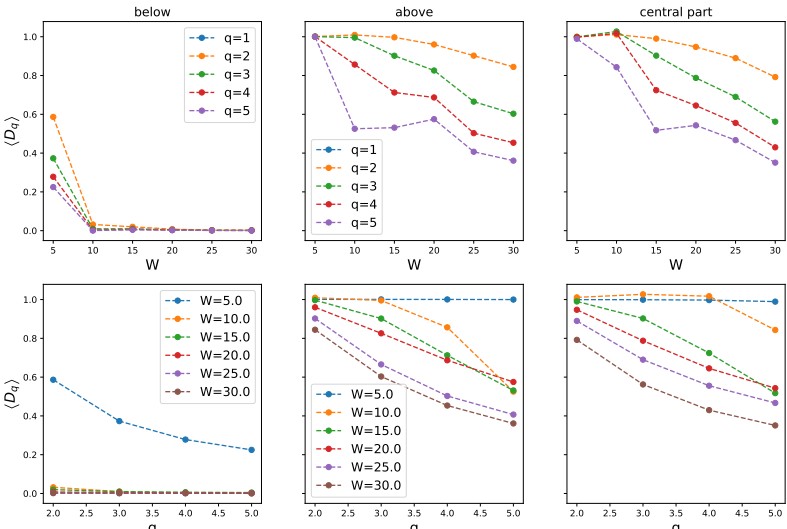

Figure 9: **Fractal dimensions $D_q$, extrapolated from the finite sizes of Fig. 7,** (upper row) versus disorder for different $q$ and (lower row) versus $q$ for different disorder amplitudes $W$ in the partially disordered RRG in different parts of spectrum (see Fig. 6): (left) below the mobility edge (localized states), $|E| > E_{ME1}$; (middle) above the mobility edge (delocalized states), $E_{ME2} \leq |E| \leq E_{ME1}$; (right) in the central (not fully ergodic) part, $|E| < E_{ME2}$. The fraction of disordered nodes is $\beta = 0.5$. Panels show gradual localization of the states below the mobility edge (left) with increasing disorder $W$ ($D_q$ goes to 0), while both states above it and at the central part stay delocalized ($D_q > 0$).

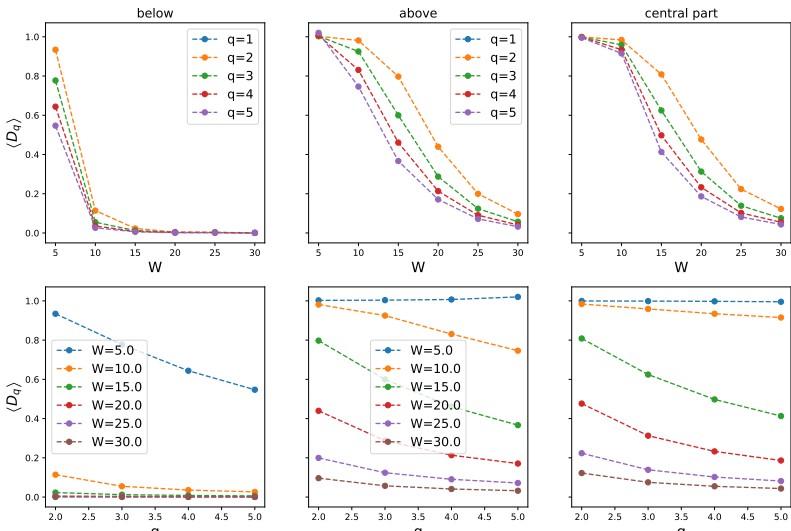

Figure 10: **Fractal dimensions $D_q$, extrapolated from the finite sizes of Fig. 8,** (upper row) versus disorder for different $q$ and (lower row) versus $q$ for different disorder amplitudes $W$ in the partially disordered RRG in different parts of spectrum (see Fig. 6): (left) below the mobility edge (localized states), $|E| > E_{ME1}$; (middle) above the mobility edge (delocalized states), $E_{ME2} \leq |E| \leq E_{ME1}$; (right) in the central (not fully ergodic) part, $|E| < E_{ME2}$. The fraction of disordered nodes is $\beta = 0.75$. Panels show gradual localization of all the states with increasing disorder $W$, both below the mobility edge, above it, and at the central part as $\beta > \beta_c = 1 - (d-1)^{-1}$, Eq. (19).

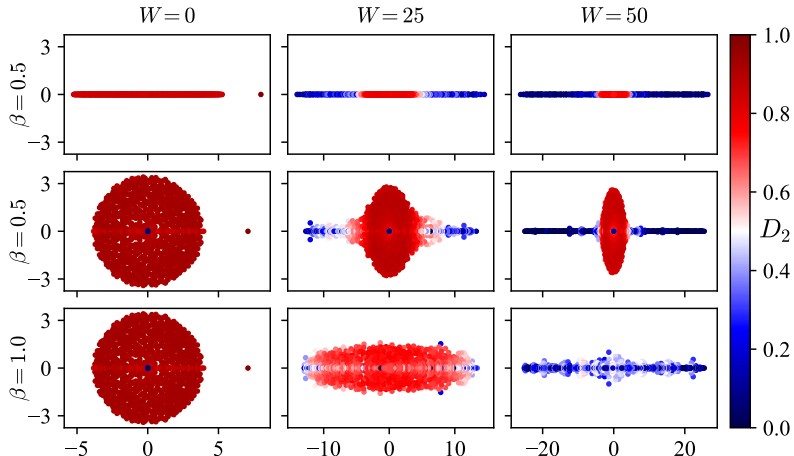

Figure 11: **Representative realizations of the complex-valued spectra in partially disordered and partially directed RRG** of the size $N = 1024$, with the connectivity $d = 8$ for different disorder strengths $W$ and the fraction $\beta$ of disordered nodes. Color coding corresponds to the fractal dimension $D_2$ of a product of left and right eigenvectors in a biorthogonal basis, $\left\langle \psi_i^L | \psi_j^R \right\rangle = \delta_{ij}$, for each point in the parameter space. The top row corresponds to an undirected Hermitian graph, $r = 1$, while the second and third ones – to the directed graphs, with the reciprocity parameter $r = 0.125$ determined as the fraction of bidirected connections to all connections. One can see that small non-Hermiticity does not break the existence of the mobility edge in the spectral central part along the real part of the energy.

# B   Further generalizations of the model

In this Appendix we consider various perturbations of the partially disordered RRG model to the directed non-reciprocal version of it [45], see Sec. B.1, and to the RRG, perturbed by the presence of short cycles of a length of 3 [22,44], which are almost absent in the standard RRG case, see Sec. B.2. In the next two subsections we investigate numerically the localization and multifractal properties of these models.

## B.1   Directed partially disordered RRG

In this section, we consider the localization in the Anderson model on a partially directed RRG with the non-Hermitian spectrum in the partially disordered case, dubbed the $\beta$-deformation of RRG. The two-parametric non-Hermitian model of RRG with standard disorder in full generality is presented in [45].

The model in [45] uses two parameters that correspond to reciprocity and hopping asymmetry. In this work only dependence on reciprocity $0 \leq r \leq 1$ is studied. A traditional way to define network reciprocity involves the ratio of the number of bidirectional connections to the number of bidirectional and unidirectional connections. We modify the RRG network as follows: with the probability $r$, we replace an undirected edge by two oppositely directed ones, with weights of 1 each. Otherwise, with probability $1 - r$, the undirected edge is changed to one directed in a random direction, with a weight of 2. Therefore, the total bandwidth of the link between connected nodes is constant and equal to 2. If $r = 0$ the graph becomes an oriented directed RRG graph, while at $r = 1$ the graph is equivalent to the standard undirected RRG. At certain ranges of parameters, this model has a tendency to become undiagonalizable due to the existence of exceptional points, see [45] for more details. To overcome the problem, small perturbation feedback $\epsilon = 2 \times 10^{-5}$ is added to unidirected edges.

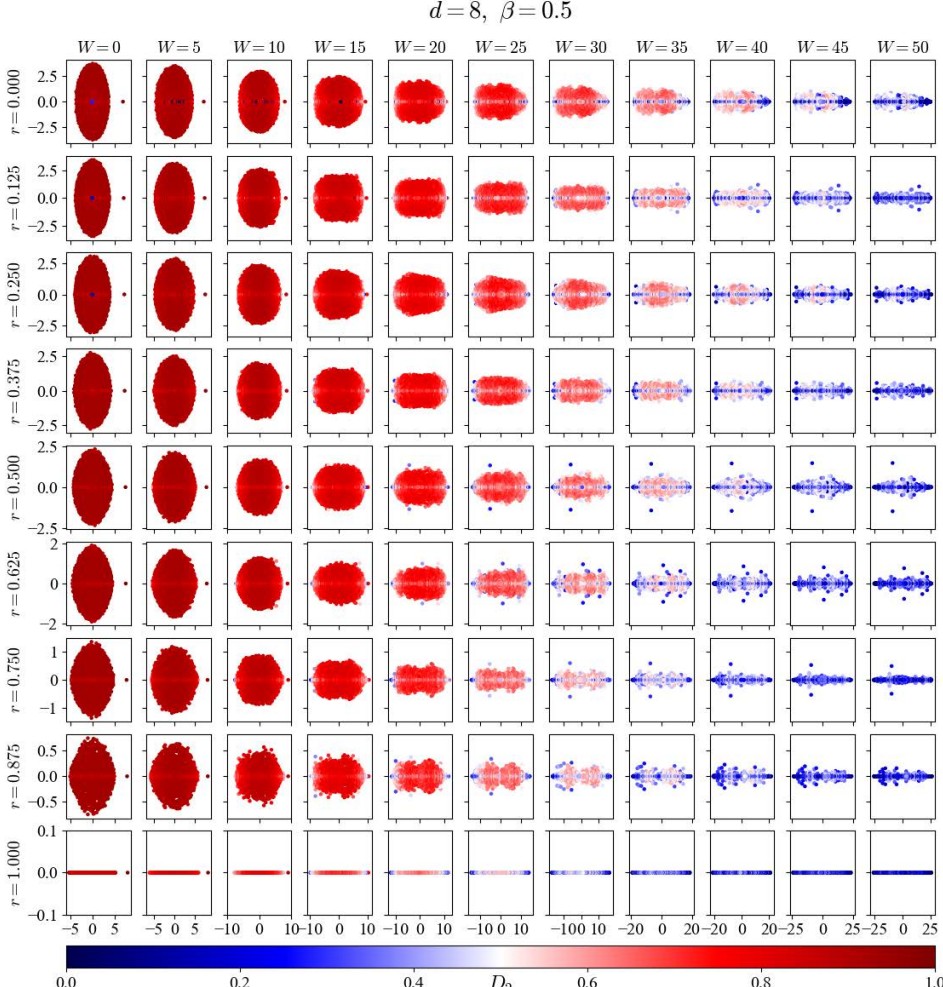

Figure 12: **Complex-valued spectra of the partially disordered and partially non-reciprocal RRG** with the node degree $d = 8$ for different reciprocity parameters $r$ and disorder amplitudes $W$ at the fraction of disordered nodes $\beta = 0.5$. Each plot is colored by the fractal dimension value $D_2$. For all panels, diagonal disorder distribution has the same realization from the interval $[-1/2; 1/2]$, but multiplied by $W$.

The representative realizations of complex-valued spectra for RRG with the connectivity $d = 8$ for different $r$, $W$, and for $\beta = 0.5$ and $\beta = 1.0$ are shown in Fig. 11. All the points in these plots are colored by the value of the fractal dimension $D_2$ of a product of left and right eigenvectors in a biorthogonal basis, $\left\langle \psi_i^L | \psi_j^R \right\rangle = \delta_{ij}$. For more details, please see Fig. 12, 13. Let us summarize the effects of competition of $\beta$ and $r$ parameters at large $W$.

- Instead of the mobility edge of the undirected case, $r = 1$, for $r < 1$, $\beta < 1$, we have the mobility curve in the complex plane. At $\beta = 0.5$ and large $W$, the spread of the imaginary parts of the delocalized states is independent of $W$. The imaginary part of the localized states at large $W$ vanishes. The latter is natural as the diagonal disorder, which is dominant, is real, see [86].

- With the parameter $r$, the width of the delocalized region along the real axis varies in the same order as the initial model.

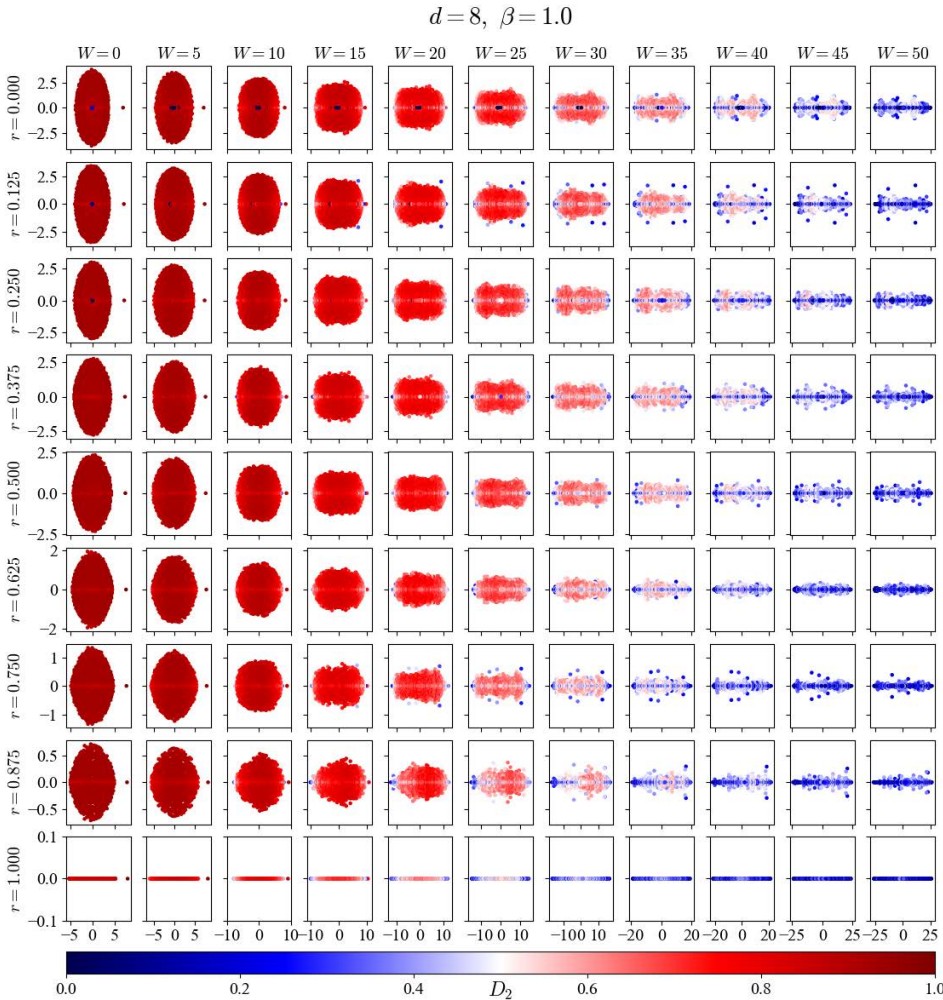

Figure 13: **Complex-valued spectra of the fully disordered ($\beta = 1$) and partially non-reciprocal RRG** with the node degree $d = 8$ for different reciprocity parameters $r$ and disorder amplitudes $W$. Each plot is colored by the fractal dimension value $D_2$. For all panels, diagonal disorder distribution has the same realization from the interval $[-1/2; 1/2]$, but multiplied by $W$.

- At $r < 1$, $\beta < 1$, the non-reciprocity leads to the emergence of the island of the localized states inside the delocalized region. Similarly to [45], this island is related to the emergence of the topologically equivalent nodes (TEN) as well as the nodes with only incoming edges (node inflows). This localized island disappears at large enough $r$.

## B.2 Effect of enhanced number of the 3-cycles

For completeness, let us consider the effect of the deformation of the RRG by a chemical potential $\mu_3$ of the 3-cycles on the localization of the partially disordered RRG. We focus on the RRG ensemble, where the degrees of all nodes are fixed to $d$ and the partition function is considered $Z(\mu_k) = \sum_{RRG} \exp(\sum_k \mu_k M_k)$, where $M_k$ is the number of the length-$k$ cycles in the graph without the back-tracking and $\mu_k$ are the chemical potentials counting the number of these $k$-cycles. Cycles of length $k$ are paths on a graph with length $k$, where all edges are different and the start and end vertex are the same.

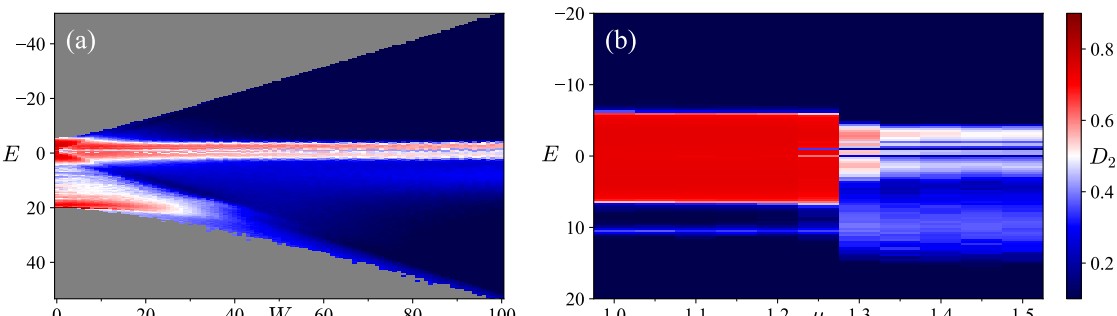

Figure 14: **Color plot of the fractal dimension $D_2$ across energy $E$ and versus (a) disorder $W$ or (b) chemical potential $\mu_3$ of 3-cycles in the partially disordered RRG of the size $N = 1024$,** with $\beta = 0.5$. Panel (a) shows the clustered phase with $\mu_3 = 2$, while panel (b) corresponds to strong disorder $W = 1000$. Each point of a color plot is averaged over totally 100 structural and disorder realizations. From both panels, one can see that the mobility edge picture survives fully in the unclustered phase, $\mu_3 < 1.3$ and at least partially even in the clusterized one.

For the $\beta = 1$, some observations concerning the localization in $\mu_3$-deformed theory can be found in [22] and a thorough analysis that uncovered quite a rich phase structure has been performed in [44] for various system sizes $N$, node degrees $d$, and cycle lengths $k$, corresponding to $\mu_k$. The number of 3-cycles can be derived from the graph adjacency matrix $M_3 \propto TrA^3$. There are four different phases in $(\mu_3, d)$ parameter space: unclustered, $\mu_3 < \mu_{3,TEN}$, TEN-scarred, $\mu_{3,TEN} < \mu_3 < \mu_c$, and two clustered ones, $\mu_3 > \mu_c, \mu_{3,TEN}$: ideal and interacting ones. At leading terms in $N$, the above critical lines are given by $\mu_{3,TEN} \sim \frac{(d-2)\ln N}{(d-1)}$ and $\mu_c \sim \frac{3(d-2)\ln N}{d(d-1)}$, see [44] for more details.

Here we shall numerically consider some effects of the $\beta$-deformation in the $\mu_3$-deformed RRG. In Figure 14(a) we present the localization pattern for fractal dimension at $\beta = 0.5$ in the $(W, E)$-plane, while in Fig. 14(b) we show its behavior in the $(\mu_3, E)$-plane.

Figure 14(b) shows the effect of $\mu_3$ on the partially disordered RRG. At small $\mu_3 < \mu_c$ in the unclustered phase, both the dependence of $D_2$ on the parameters and the position remain the same as in Figure 1. At $\mu_3 > \mu_c$ (see Fig. 14(b) at $\mu_3 > 1.3$), the system undergoes the clusterization transition [44]. The $W$-dependence of $D_2$ changes, as shown in Fig. 14(a). The localization in the $\beta$-deformed RRG model occurs in each cluster separately. This effectively replaces $N$ by $d + 1$ and suppresses $D_q$ value in Fig. 14(b). In the clustered phase, the center of the continuous spectrum part shifts to $-1$ because the graph consists of dense clusters of triangles and narrows due to the change of DOS from KM distribution to triangular shape distribution, like in the diagonal disorder-free case.

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
