# Peer review of "Robust extended states in Anderson model on partially disordered random regular graphs"

_SciPost Physics, doi:SciPost Phys. 16, 106 (2024)_

## Round 1 · Referee Report · Anonymous (Referee 1) · 2023-11-1

Strengths

1-The model under investigation is relevant and the physical motivation is compelling

Weaknesses

1- The language and presentation of the paper is very confusing 2- The size of the numerical model with N=1024 seems a bit low, especially considering the discrepancy in Fig.3 c) d)

Report

The article discusses the survival of delocalized states of free particles on a random regular graph where a fraction of nodes are subject to a strongly disordered potential. This is a relevant problem, since this model can serve as a toy model for the Hilbert space for certain interacting systems.
However, this paper suffers from poor presentation making it at times very hard to understand what the authors want to convey.

Question to the authors: If we take the W → ∞ limit like it seems to be done in section 3, does the model reduce to the free problem on the non-disordered subset of nodes (and some isolated disordered states)? Concretely this would be a RRG where each node has a β probability of being removed, or (at least in the case of sparse graphs) a random graph with connectivity distributed according to Eq. (7). This reduction seems rather obvious to me, but maybe I overlooked something. If this is indeed the case it should be stated clearly and not hidden in the derivation on the top of page 7. I cannot judge if this problem has been studied elsewhere before.

The abstract should be overhauled, specifically it would be good to write the meaning of (β,d) in one sentence rather then spread it out over three. The last sentence is very confusing to me.
Section 1 introduce the topic and motivate the model. The physical motivation in the introduction, namely the study of interacting models with topologically protected modes, is sound. However it could benefit from some references. Furthermore, there is an existing body of research on models comprised of clean and disordered parts (for example PHYSICAL REVIEW X 9, 041014 (2019), but there is a lot more). It would be helpful to clarify the relationship between those works and the present work.
Section 2 defines the model and demonstrate the survival of extended states numerically. Fig. 1 is convincing, however an explanation of the additional branch starting at E=20 would be welcome. Furthermore, a comment on numerical convergence with respect to all relevant parameters is missing.
Section 3 focuses on deriving a formula for the density of extended states and the percolation threshold β(d) in the W →∞. It is described that the disordered nodes do not contribute in the limit W→∞ (c.f. my question above). Eq. (17) is derived under the assumption that G is self averaging. Can you comment on why this holds?
Fig.2, why does the purple curve deviate from the blue dotted curve? can you comment on the choice of parameters? Convergence with sample and model size?
Fig. 3 c) and d) show clear deviations from the predicted curve, in particular there are a considerable number of states outside of the predicted band and the distribution looks more pointy then the predicted distribution. Why is that?
Fig 4, what does the grey color signify? If it means that there are no states with this energy, doesn't the black curve signify the band edge rather then the mobility edge? What is that extra branch in panel a)?
Section 4 discusses how RRG with connectivity d relate to RRG with connectivity N-d-1. I fail to see how this section connects to the rest of this work.
Section 5 discusses further generalizations of this model, in particular directed graphs and chemical potentials. I cannot see the relation to the previous part of the paper nor to the physical motivation mentioned in the introduction. Both sections introduce a significant amount of new concepts and terminology while adding very little to the story of the paper. Maybe it would be better to spin them off in a different work?
The captions of Fig. 6 and 7 are insufficient.
The numerical results plots look compelling, however no comment is given on convergence with respect to system size and sample count. In its current form the article is very difficult to follow due to its presentation; I can therefore not recommend publishing without major revisions to language and presentation.

Requested changes

1- Significantly improve the language and presentation. Here just a couple points; this list is not exhaustive: - The abstract is hard to parse and should be reformulated, especially the last sentence. - (d,β) and (β,d) are used inconsistently - some concepts are not introduced like "short-cycles" - grammatical mistakes are frequent - Captions only state what was plotted but not what one should look at - Sentences are at times very long and cover different only loosely related ideas - 2- add references to the physical motivation as well as compare to other works dealing with models comprising clean and disordered parts. 3- Comment on convergence of numerical computations with respect to relevant parameters like number of samples, number of nodes, etc. The number of nodes N=1024 seems very low to me. Fig 3 c) and d) shows disagreement between observed and predicted density of states, yet there is no comment on this discrepancy anywhere. 4- Explain the second branch in Figure 1 5- Either integrate section 4 and 5 better with the rest of the text or split them off in a different work. 6- Section 3 seems to be about the large disorder limit, clarify this

  • validity: ok
  • significance: good
  • originality: ok
  • clarity: low
  • formatting: below threshold
  • grammar: mediocre

Author:  Ivan Khaymovich  on 2023-12-03  [id 4166]

(in reply to Report 1 on 2023-11-01)

Dear Referee 1,

We are grateful for your careful reading of the manuscript and useful comments.
Please see attached the reply, followed by the revised text with the changes, highlighted by red font.

Sincerely yours,
the authors.

Attachment:

Reply1red_text.pdf

---

## Round 1 · Referee Report · Anonymous (Referee 2) · 2023-11-6

Strengths

See my report

Weaknesses

See my report

Report

The article discusses the recently discovered localization transition in partially disordered random regular graphs (as cited in Ref. [34]). The Anderson transition in random graphs with effective infinite dimensionality has garnered significant interest recently, partly due to its analogy with many-body localization. Recent studies have uncovered intriguing properties, particularly the potential for a non-ergodic delocalized phase. In this paper, the authors investigate partial disorder, where a fraction of the random graph sites has zero disorder. Their primary objective is to elucidate the mechanism behind the presence of a mobility edge, regardless of the disorder strength, leading to the delocalization of states near the middle of the band. The article first describes this aspect and then extends the study to related cases, such as the directed/non-Hermitian scenario and the introduction of a chemical potential for 3-cycles.

While I find the paper's subject matter interesting, I have several reservations about its current form, which prevent me from recommending it for publication in Scipost Phys. I have outlined my comments below:

  1. Understanding the Mobility Edge Mechanism: The initial part of the paper appears to be the most crucial. It delves into the mechanism behind the mobility edge in the presence of partial disorder. Let me rephrase the argument to confirm my understanding: the authors explore the limit of infinite disorder and formulate Abou-Chacra-Thouless-Anderson recursion relations for the cavity Green's function in the clean (non-disordered) region. Essentially, infinite disorder excludes certain neighbors, causing fluctuating connectivity between clean sites. The authors approximate this problem by neglecting connectivity fluctuations, justified in the limit of a large connectivity. This approach leads to a self-consistent equation for the cavity Green's function, allowing the prediction of the mobility edge.

It appears to me that this problem resembles the recent rigorous solution for the localization/delocalization transition of eigenstates of the Adjacency matrix (where onsite potentials are zero) of Erdös-Rényi graphs, as detailed in arXiv:2005.14180, arXiv:2305.16294. Notably, states can be localized due to fluctuating connectivity. However, I find it unclear how this localization is described in the authors' self-consistent approach, where connectivity fluctuations are neglected.

I have additional questions concerning the numerical simulations. Figure 4 seems crucial, demonstrating the good agreement between the analytical formula for the mobility edge (mentioned only in the caption, not in the main text?) and the numerical data. However, the plotted variable is D2​ as a function of the parameters E and β. The color scale abruptly changes at D2​=0.5 (white), with mainly red for D2​>0.5 and blue for D2​<0.5. I fail to understand why D2​ is set to 0.5 at the transition. On the contrary, I would expect D2​ to tend to 0 slowly with size. My query is: since the authors possess a theoretical formula, can they precisely determine the transition numerically? The disorder does not necessarily need to be W=1000, and the section where the entire band is delocalized might not be as relevant. The focus should be on the localization transition, which is the crucial aspect of interest.

  1. Sparse and Dense RRG Duality: The subsequent section briefly outlines a duality between sparse and dense Random Regular Graphs (RRG). I struggle to comprehend the authors' motivation for considering this case. Additionally, the techniques employed lack sufficient explanation for my understanding. Numerical simulations in this limit must be notably challenging, likely constrained by a smaller system size.

  2. Generalizations of the Model: The other two model generalizations are quite challenging to grasp. The directed couplings case is excessively elusive, referring to a future publication. It is difficult to discern the message and the link between these results and the previous sections. The explanation of the 3-cycles case is insufficient. What exactly are 3-cycles? Why have they been included in the study?

  3. Figures in the Appendix: There are numerous figures in the appendix, and their purpose is unclear to me. Could the authors provide context or explanations for these figures?

  4. Language and Text Quality: Lastly, the English and overall text quality require careful editing. Several excellent tools are available to accomplish this.

Requested changes

See my report

  • validity: ok
  • significance: ok
  • originality: good
  • clarity: ok
  • formatting: acceptable
  • grammar: acceptable

Author:  Ivan Khaymovich  on 2023-12-03  [id 4165]

(in reply to Report 2 on 2023-11-06)
Category:
answer to question

Dear Referee 2,

We are grateful for your careful reading of the manuscript and constructive critique.
Please see attached the reply, followed by the revised text with the changes, highlighted by red font.

Sincerely yours,
the authors.

Attachment:

Reply2red_text.pdf

---

## Round 2 · Referee Report · Anonymous · 2023-12-6

Report

I thank the authors for addressing my scientific questions and agree that the
results merit publication in SciPost Physics. However, the presentation of
those results still can and should be improved.

In particular:

New Figure 2:
The authors added another Figure to the manuscript which illustrates the
fractal dimension (panel a)) and the density of states (panel b)) at moderate
disorder strength W=30. I believe this addition is useful, as it connects the
article to finite disorder strengths, complementing the previous focus on the
W→∞ limit. Do I understand correctly that the authors assume that the localized
states around E=0 in Fig. 5, central peak in the density of states in Fig. 2b)
and 4c)d) as well as the dip in the fractal dimension in Fig. 2a) are due to
isolated clean nodes and pairs of clean nodes? Something to this effect is
stated at the bottom of page 9, but it should probably be stated right were
Fig. 2 is discussed in the text. Instead the text is currently very vague about
these deviations, even though the black (theoretical) curve strongly differs
from the red (numerical) curve in panel b). The text describing Fig. 4 is
similarly vague. Furthermore, there is no explanation of the N→∞ extrapolation
in Panel a).

Section 4:
I can now follow the motivation of including the discussion on the duality
between sparse and dense RRG. However, the first couple sentences of this
sections are very difficult to understand. What do you mean by "bare degree d"?

Abstract:
I find the abstract now easier to understand. In the first sentence, I
recommend omitting the subclause ", the location of which is under control."
since it sounds awkward and seems unnecessary. The last sentence is still
hard to understand and contains grammatical errors. Maybe splitting it up and
giving more context would help.

Grammar and style:
The manuscript still contains frequent grammatical mistakes and awkward
phrasings. For example, in the first two sentences of the main text, there are
two missing definite articles. These mistakes distract the reader from the
scientific content. As the other referee pointed out, nowadays there are plenty
of tools widely available that can point out grammatical and stylistic
problems.

  • validity: ok
  • significance: good
  • originality: good
  • clarity: low
  • formatting: acceptable
  • grammar: below threshold

Author:  Ivan Khaymovich  on 2023-12-10  [id 4180]

(in reply to Report 1 on 2023-12-06)
Category:
answer to question

We thank the referee for thorough and quickly submitted referee report.

Please see our reply with the updated manuscript version with the red-colored main changes.

Attachment:

Reply1_stage2red_text.pdf

---

## Round 2 · Referee Report · Anonymous · 2024-1-9

Strengths

see my previous report

Weaknesses

see my report

Report

The main scientific question I raised in my previous report concerned the localization effects induced by fluctuating connectivity. I must express my dissatisfaction with the authors' response on this matter. It is well-established that fluctuating connectivity can induce localization of eigenstates in sparse adjacency matrices, as evidenced in Ref. G. Biroli and R. Monasson, J. Phys. A: Math. Gen. 32, L255 (1999), where it is stated that "localized eigenvectors are centered on geometrical defects, that is on sites whose number of neighbors is much smaller or much larger than the average connectivity." The condition for such an effect is not too stringent, given approximately by |q-c| > √q, where q is the anomalous connectivity of the site and c is the average connectivity.

It appears to me that for small enough beta, this effect should play a role. Hence, the rigorous arguments presented in Refs. [65,66] should be relevant here, appropriately adapted to the present context. The recent work by M. Tarzia, PHYSICAL REVIEW B 105, 174201 (2022), is also pertinent, in my opinion. Essentially, the authors make two approximations for d << N: infinite disorder and fixed connectivity d*. A precise numerical study of the effects of finite disorder and fluctuating connectivity is necessary to explain the deviations observed between the numerical data and the analytic predictions, especially at small beta.

The case of a dense graph with d ≈ N is less clear. The mapping of such a Random Regular Graph (RRG) to a Cayley tree should not be valid in this limit, so it is not surprising that a direct use of the previous cavity approach does not work. I still believe the authors could provide more details about their approach for this part to be clearer.

In my previous report, as well as that of the other referee, I highlighted formulations that were difficult to understand or grammatically incorrect. The corrections made by the authors do not entirely satisfy me. The abstract, as well as various passages in the articles, are still not well-written in my opinion. I provide below some examples, which are not exhaustive:

-The abstract states that the authors deal with the "ensemble of random regular graphs (RRG), with the connectivity d and the fraction β of disordered nodes." I would suggest stating that the authors consider partially disordered random regular graphs, i.e., with a fraction β of the sites being disordered, while the rest remain clean. The authors study the influence of the connectivity d and fraction β on the localization/delocalization properties of the states.
- The last sentence about the perturbations does not seem necessary to include in the abstract, as the material has been placed in the appendix. In any case, I think the authors should change the "non-reciprocity parameter for edges" to non-Hermitian directed hoppings (non-reciprocity is not clear to me).
- In the caption of Fig. 5, "Black solid (dashed) line denotes the mobility edge, |E_{M E} |^2 = 4(1 − β)(d − 1), Eq. (19), (|E_{M E} + 1|^2 =4(1 − β)(d − 1))." can be confusing: Firstly, the equation is not (19) but (18), and this refers to the case described in (a) while it should also be stated that the parenthesis refers to the case (b).
- Still in this caption of Fig. 5, the panel (c) corresponds to "the central band |E| < E_{ME}": this information is not sufficient to understand what is represented. Is what is plotted D_2 at E=0 or an average of D_2 over the central band? Eq. (19) gives 1-beta_c=1/(d-1) but for RRG it should be 1/d instead, so what was used in Fig. 5 (a)?
-The end of the first paragraph in section 4 is still confusing: "but for any values of the total vertex degree d of the graph." d* < d and d* >> 1, therefore d >> 1.
-I don't think the expression "complimentary graph" is the right one; rather, it should be "complementary graph."
-d_eff just before the conclusion should be written as d_\text{eff}.
-In the color plots, I still believe that a Perceptually Uniform Color Map should be used instead of the color map chosen by the authors. As I mentioned earlier, the chosen color map emphasizes D_2=0.5, which is not particularly distinctive in their data.
- In Fig. 4, the authors should indicate that the black line corresponds to the generalized Kesten-McKay formula.
- Below Eq. (13), the relative fluctuations sigma_d/d* = sqrt(beta)/sqrt((1-beta)d) are not small for small beta, even at d=10 or 20.
-In Eq. (19), either write the left equation or the one on the right side of the arrow. For RRG it should be d in the denominator, not d-1.

Once again, this is not an exhaustive list. Moreover, there is a significant amount of formulations that are not typical English. This can be polished very easily using, for example, ChatGPT or another such tool (which I have used to polish my report).

Requested changes

Citations for Fluctuating Connectivity and Localization Effects:
Kindly include appropriate references to prior works discussing the localization effects induced by fluctuating connectivity. A relevant study is provided by G. Biroli and R. Monasson in J. Phys. A: Math. Gen. 32, L255 (1999). Additionally, cite studies that have numerically investigated these effects in conjunction with finite disorder, such as the work by M. Tarzia in PHYSICAL REVIEW B 105, 174201 (2022).

Editing the Manuscript:
Requesting a thorough review and editing of the manuscript in accordance with the feedback provided in my report. Key areas of focus include refining the abstract to accurately describe the study as focusing on 'partially disordered random regular graphs.' Ensure that figure captions are clear, equations are correctly referenced, and overall grammatical issues are addressed for enhanced readability.

  • validity: good
  • significance: good
  • originality: good
  • clarity: good
  • formatting: reasonable
  • grammar: acceptable

Author:  Ivan Khaymovich  on 2024-02-19  [id 4316]

(in reply to Report 2 on 2024-01-09)
Category:
answer to question

We thank the referee for thorough submitted referee report.

Please see our detailed reply with the updated manuscript version with the red-colored main changes.

Attachment:

Reply2main_.pdf

---

## Round 2 · Author Response

Dear Editor,

We have addressed all Referees' questions and comments.
Please see the replies to both referees in the previous version, followed by the manuscript, where changes are highlighted with red font.

Thus, we resubmit our manuscript to SciPost Physics and believe that our work meets all the criteria of the journal.

Sincerely yours,
the authors.

---

## Round 2 · List of Changes

1 - the abstract has been reformulated to make it clearer.
2 - in the introduction, the references on MBL and typically protected modes there have been added with the corresponding discussion.
3 - the motivation of the generalisations of the model has been clarified in the end of the introduction.
4 - all the figure captions have been extended to make it clearer.
5 - an additional energy level, starting at $E=d$, has been described in the text.
6 - Fig. 2 clarifying the finite-size effects up to $N=3 \cdot 10^4$ with the corresponding discussion has been added to the text.
7 - some deviations from Poisson statistics in Fig. 3 have been discussed in the text.
8 - the contribution to the density of states (DOS) from the localized states, barely seen in Figs. 4 and 5(a), has been discussed in detail and shown in Fig. 3(b) at intermediate disorder values.
9 - the qualitative summary of the analytical results, based on the graph of clean nodes, has been added to Sec. 3.
10 - the self-averaging property of the Green's function at large effective connectivity has been discussed before Eq. (14).
11 - the mobility edge has been explicitly added as Eq. (18).
12 - some discussions of possible origins of DOS deviations from the analytical predictions have been added to the end of Sec. 3.
13 - the motivation of the generalisations of the model and the robustness of the mobility edge to perturbations has been added to the end of Sec. 3 and the beginning of Sec. 4.
14 - the references [65, 66] on the localisation in Erdos-Renyi graph due to the focusing of node degree have been added and their relations to our results have been discussed.
15 - Sec. 5 has been moved to the Appendix B and restructured, including some figures.
16 - discussions in the Conclusion have been slightly reformulated.
17 - Discussions in Appendix A have been extended and clarified.

---

## Editorial Decision

published